# Cortical plasticity is associated with blood–brain barrier modulation

Evyatar Swissa[1], Uri Monsonego[2], Lynn T Yang[3,4], Lior Schori[2], Lyna Kamintsky[5], Sheida Mirloo[5], Itamar Burger[2], Sarit Uzzan[6], Rishi Patel[3], Peter H Sudmant[3], Ofer Prager[1,2], Daniela Kaufer[3,4], Alon Friedman[1,2,5]*

[1]Department of Brain and Cognitive Sciences, The School of Brain Sciences and Cognition, Zlotowski Center for Neuroscience, Ben-Gurion University of the Negev, Beer-Sheva, Israel; [2]Department of Physiology and Cell Biology, Faculty of Health Sciences, Ben-Gurion University of the Negev, Beer-Sheva, Israel; [3]Department of Integrative Biology, University of California, Berkeley, Berkeley, United States; [4]Helen Wills Neuroscience Institute, University of California, Berkeley, Berkeley, United States; [5]Department of Medical Neuroscience, Dalhousie University, Halifax, Canada; [6]Department of Clinical Biochemistry and Pharmacology, Faculty of Health Sciences, Ben-Gurion University of the Negev, Beer-Sheva, Israel

*For correspondence:
Alon.Friedman@dal.ca

Competing interest: The authors declare that no competing interests exist.

**Abstract** Brain microvessels possess the unique properties of a blood–brain barrier (BBB), tightly regulating the passage of molecules from the blood to the brain neuropil and vice versa. In models of brain injury, BBB dysfunction and the associated leakage of serum albumin to the neuropil have been shown to induce pathological plasticity, neuronal hyper-excitability, and seizures. The effect of neuronal activity on BBB function and whether it plays a role in plasticity in the healthy brain remain unclear. Here we show that neuronal activity induces modulation of microvascular permeability in the healthy brain and that it has a role in local network reorganization. Combining simultaneous electrophysiological recording and vascular imaging with transcriptomic analysis in rats, and functional and BBB-mapping MRI in human subjects, we show that prolonged stimulation of the limb induces a focal increase in BBB permeability in the corresponding somatosensory cortex that is associated with long-term synaptic plasticity. We further show that the increased microvascular permeability depends on neuronal activity and involves caveolae-mediated transcytosis and transforming growth factor β signaling. Our results reveal a role of BBB modulation in cortical plasticity in the healthy brain, highlighting the importance of neurovascular interactions for sensory experience and learning.

## eLife assessment

This study builds upon previous work which demonstrated that brain injury results in the entry of a protein called albumin into the brain which then causes diverse effects. The present study shows that prolonged stimulation of a forelimb in a rat leads to albumin entry and is associated with effects that suggest plasticity is enhanced in the stimulated side of the brain. The strength of evidence was **convincing** and results are **important** because they suggest a previously considered pathological process may be relevant to the normal brain and have benefits.

## Introduction

The intricate components comprising the blood–brain barrier (BBB) regulate the bi-directional transport of molecules between the brain and the circulatory system and maintain the extracellular environment needed for normal function of the neurovascular unit (NVU) (*Chow and Gu, 2015*). Dysfunction

of the BBB is common in many neurological disorders and is associated with impaired cross-BBB influx and efflux (*Obermeier et al., 2013*). For example, seizures were shown to increase cross-BBB influx of serum albumin in a process partially mediated by excessive release of neuronal glutamate and subsequent activation of *N*-methyl-ᴅ-aspartate (NMDA) receptors (*Vazana et al., 2016*).

Emerging evidence suggests that cross-BBB influx/efflux may also change in response to physiological neuronal activity in the healthy brain: (1) whisker stimulation (1 hr, 2 Hz) was shown to increase BBB influx of insulin-like growth factor-I in rats (*Nishijima et al., 2010*); (2) physiological neuronal activity was shown to regulate BBB efflux transport in mice (*Pulido et al., 2020*); and (3) circadian rhythms were shown to alter BBB efflux in *Drosophila* (*Zhang et al., 2018*) and mice (*Pulido et al., 2020*). However, the mechanisms and the role of these physiological changes in BBB properties are not known.

Here, we sought to test whether physiological BBB modulation may involve mechanisms previously linked to pathological BBB changes, for example, transcytosis of serum albumin (*Ivens et al., 2007*; *Seiffert et al., 2004*) and transforming growth factor β (TGF-β) signaling (*David et al., 2009*; *Heinemann et al., 2012*; *Ivens et al., 2007*; *Weissberg et al., 2015*). As TGF-β signaling has been linked to synaptic plasticity (*Dahlmanns et al., 2023*; *Gradari et al., 2021*; *Patel and Weaver, 2021*; *Weissberg et al., 2015*), we also explored the hypothesis that physiological BBB modulation may be associated with neuroplasticity. Lastly, we sought to provide the first evidence for physiological BBB modulation in humans.

## Results

### Stimulation increases BBB permeability

To test for changes in BBB permeability in response to stimulation, we combined electrophysiological recordings with BBB imaging using intravital microscopy (*Figure 1*, see timeline of stimuli in *Figure 1a*). First, we localized vascular response to sensory stimulation using wide-field imaging of arterial diameter and monitoring change in total hemoglobin (HbT) signal during a 1 min stimulation of the limb (*Figure 1b–d*; *Bouchard et al., 2009*). Fluorescent angiography before and immediately after 30 min of limb stimulation showed local extravasation of the tracer sodium fluorescein (NaFlu) around the responding blood vessels, indicative of BBB leakage (*Figure 1f and g*). Histological analysis confirmed the presence of NaFlu in the extravascular space around small vessels in the somatosensory cortex, contralateral (and not ipsilateral) to the stimulated limb (*Figure 1m and n*). Next, we intravenously injected a separate cohort of rats with either the albumin-binding dye Evans blue (EB) or albumin conjugated with Alexa Fluor 488 (Alexa488-Alb) without performing craniotomy. Brain extravasation of both dyes was found in the contralateral hemisphere, indicating the presence of serum albumin in the neuropil (*Figure 1h–k*). ELISA performed in cortical tissue at different time points after stimulation confirmed a higher concentration of albumin in the stimulated sensorimotor cortex early after stimulation (30 min) compared to non-stimulated sham controls, which declined 4 and 24 hr post-stimulation (*Figure 1l*). As a positive control, brain tissues were also prepared from rats subjected to photothrombosis (PT)-induced ischemia (see 'Materials and methods'). Albumin concentrations in the peri-ischemic cortex 24 hr following PT were significantly higher compared to sham and all time points after stimulation (*Figure 1—figure supplement 1d*). These results suggest that the focal modulation of BBB permeability following stimulation is distinct from the extensive BBB dysfunction observed in injury models.

### BBB permeability is associated with synaptic potentiation

To study the changes in brain activity during repetitive stimulation of the limb, we measured somatosensory-evoked potentials (SEPs) by averaging neuronal response to a 1 min test stimulation before and after the 30 min stimulation train (6 Hz, 2 mA). SEP amplitudes and area under the curve were increased in stimulated, but not in control cortices (*Figure 2a–e*). Potentiation lasted until the end of each experiment (*Figure 2f*) and was therefore indicative of long-term potentiation (LTP).

To test whether stimulation-induced albumin extravasation has a role in LTP, we recorded local field potentials (LFPs) before and after exposure of the cortex to 0.1 mM albumin (corresponding to 25% of serum concentration) (*Ivens et al., 2007*). The relatively high concentration of albumin was chosen to account for factors that lower its effective tissue concentration, such as its low diffusion rate and

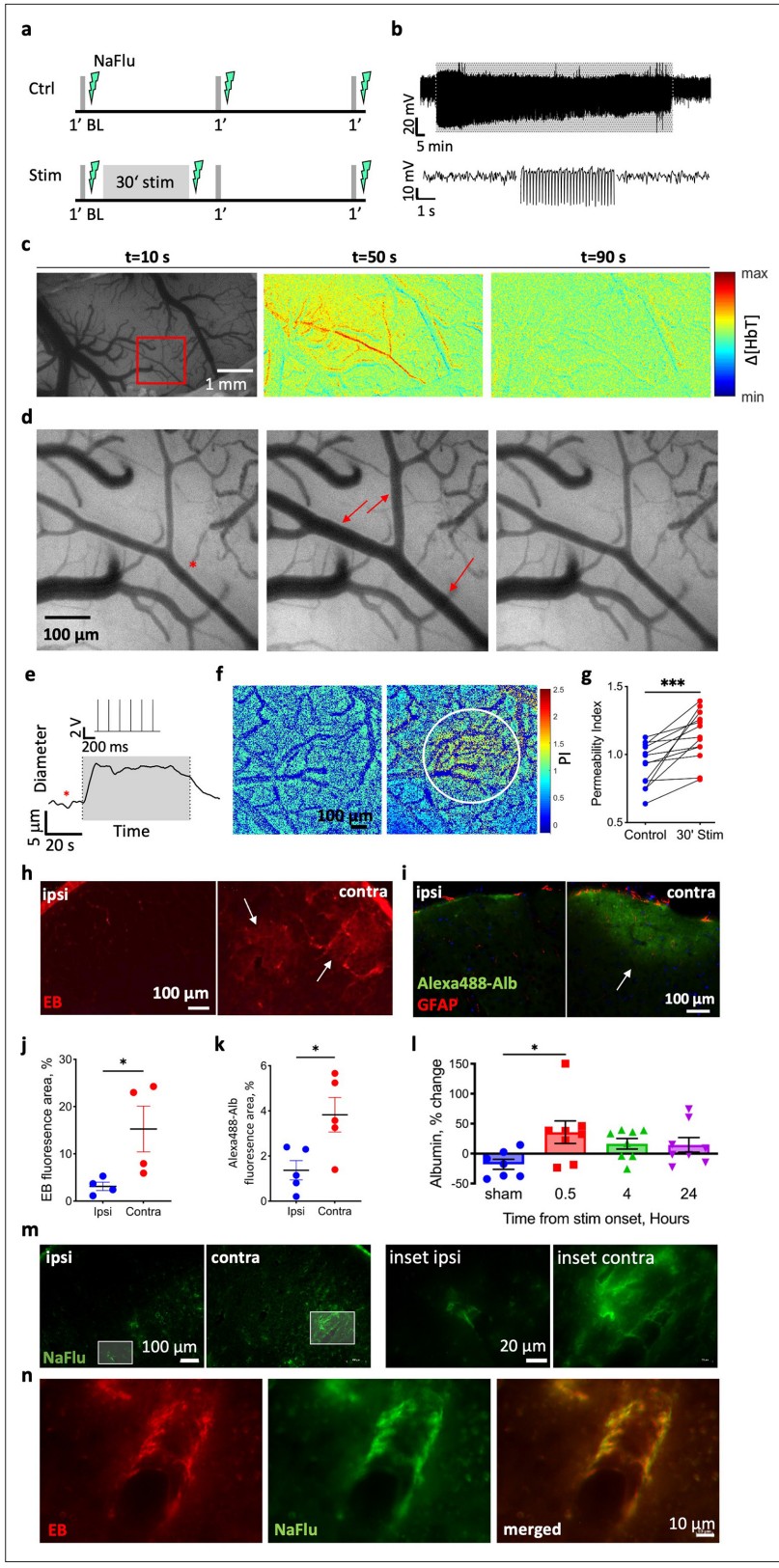

**Figure 1.** Limb stimulation modulates blood–brain barrier (BBB) permeability. (**a**) The experimental paradigm in control animals (Ctrl, top panel) and animals that underwent a 30 min stimulation (Stim, bottom panel). Gray lines indicate 1 min test stimulations, lightning indicates the timing of BBB imaging (using injection of sodium fluorescein, NaFlu). (**b**) Top: local field potential (LFP) trace from the corresponding region in L2/3 sensorimotor

*Figure 1 continued on next page*

*Figure 1 continued*

cortex before, during (grayed area), and after 30 min limb stimulation. Bottom: 5 s excerpts of the above trace (left-to-right: before during and after stimulation). (**c**) Image of the cortical window over the rat sensorimotor cortex, followed by the change in total hemoglobin (Δ[HbT]) concentration maps showing the evolution of the hemodynamic response during and after stimulation (t states the time point of the image; 120 s total: 30 s before and after a 60 s stimulation). Red rectangle marks the responding region magnified in (**d**). (**d**) Images of the responding arteriole (dilated parts marked with red arrows) in the rat's cortex before (left) during (middle) and after stimulation (right). The red asterisk denotes the measured arteriole on panel (**e**). (**e**) The diameter of the responding arteriole during 1 min stimulation. Gray area corresponds to time of stimulation (the stimulation trace is shown above the grayed area). (**f**) NaFlu permeability maps before (left) and after 30 min stimulation (right) showing tracer accumulation around a responding arteriole (marked with a white circle). (**g**) Permeability index (see 'Materials and methods', under in vivo imaging) is higher after stimulation compared to baseline (n = 13 rats, mean ± SEM, Wilcoxon, p<0.001). (**h, i**) Fluorescence images of cortical sections of stimulated rats injected with the albumin-binding dye Evans blue (EB) (**h**) and Alexa-488-Alb (**i**). White arrows point to areas with tracer accumulation. (**j, k**) Total fluorescence of EB (**j**) and Alexa488-Alb (**k**) after stimulation in the contralateral hemisphere compared to the ipsilateral hemisphere. (EB n = 4 rats, 32 sections, nested *t*-test, p=0.0296; Alexa488-Alb n = 5 rats, 20 sections, nested *t*-test, p=0.0229; mean ± SEM). (**l**) Albumin concentration in the contralateral hemisphere relative to the ipsilateral in three different time points after stimulation compared to sham stimulation. (0.5, 4 and 24 hr post-stimulation n = 8, sham n = 7, mean ± SEM, Kruskal–Wallis with false discovery rate (FDR) correction, p=0.0242, q = 0.0406). (**m**) Cortical sections of the area of limb representation from both hemispheres of a stimulated rat (left) and higher magnification images (right). (**n**) In an animal injected with both EB and NaFlu post stimulation, fluorescence imaging shows extravascular accumulation of both tracers along a cortical small vessel in the stimulated hemisphere. *p<0.05, ***p<0.001.

The online version of this article includes the following source data and figure supplement(s) for figure 1:

**Source data 1.** All data measured and analyzed for *Figure 1*.

**Figure supplement 1.** Increased blood–brain barrier (BBB) permeability following stimulation.

**Figure supplement 1—source data 1.** All data measured and analyzed for *Figure 1—figure supplement 1*.

---

its likelihood to encounter a degradation site or a cross-BBB efflux transporter (*Tao and Nicholson, 1996*; *Zhang and Pardridge, 2001*).

Albumin application increased the amplitude and frequency of spontaneous activity (*Figure 2g and k*). Neuronal response to the stimulation showed synaptic potentiation in the presence of albumin compared to that recorded under aCSF (*Figure 2h–j*), indicating that cortical exposure to serum albumin results in synaptic potentiation.

## BBB modulation is activity-dependent

To test the neurotransmission pathways involved in the regulation of BBB permeability during stimulation, we exposed the cortical window to selective antagonists of ionotropic glutamate receptors (*Figure 3a*). The AMPA/kainate receptor blocker CNQX prevented the stimulation-induced BBB modulation, whereas the NMDAR blocker AP5 did not (*Figure 3b and c*). CNQX also reversibly blocked synaptic activity and prevented the vascular response to neuronal activation, indicated by the absence of vasodilation (AKA 'neurovascular coupling [NVC]'; *Figure 3d and e*, *Figure 3—figure supplements 1a* and *2a and c*). These results indicate that stimulation-induced BBB modulation was mediated by neuronal activation. In contrast, AP5 did not affect baseline SEP (*Figure 3d and e*), nor the vascular response (*Figure 3—figure supplements 1a* and *2b*). Synaptic potentiation was blocked under both CNQX or AP5 (*Figure 3d, e and g, h*, *Figure 3—figure supplement 1a*), consistent with the crucial role of AMPA and NMDA receptors in synaptic plasticity. Next, we tested for molecular correlates of synaptic strength by comparing postsynaptic density 95 (PSD-95) expression levels in the cortex at different time points. PSD-95 is known to increase following LTP and AMPAR trafficking (*El-Husseini et al., 2000*; *Holtmaat and Svoboda, 2009*). Western blot analysis showed significantly higher levels of PSD-95 expression in stimulated rats as compared to sham at both 30 min and 24 hr after stimulation (*Figure 3—figure supplement 1c and d*).

## Albumin extravasation and LTP require caveolae-mediated transcytosis

In endothelial cells, albumin is transported mainly via caveolae-mediated transcytosis (CMT) through binding to the gp60 receptor (*Tiruppathi et al., 1997*). To test whether CMT mediates

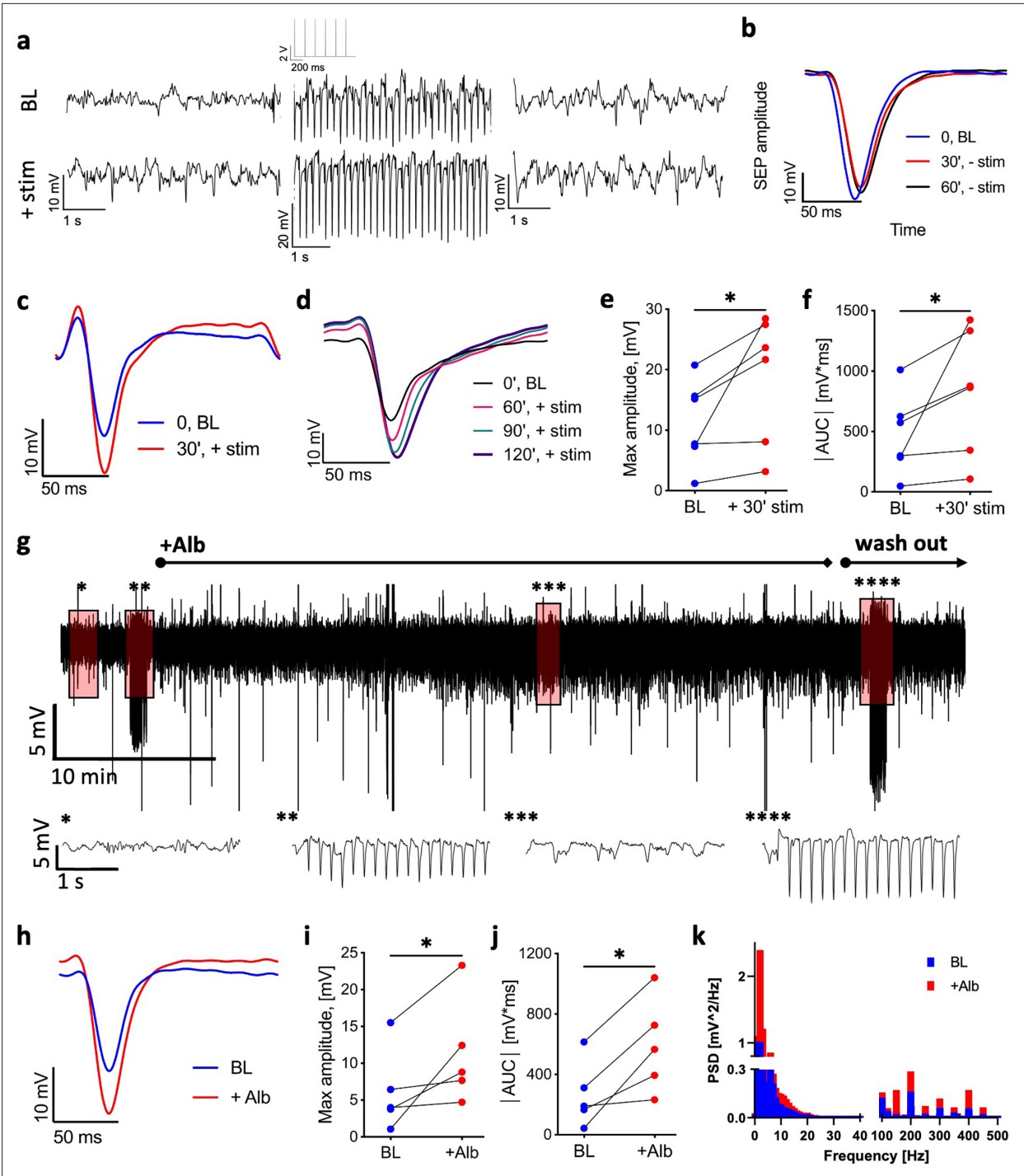

**Figure 2.** Stimulation and cortical perfusion of serum albumin induce long-term potentiation (LTP). (**a**) Top: LFP trace from the rat L2/3 sensorimotor cortex before (left), during (middle), and after (right) test stimulation (1 min, 6 Hz, 2 mA, an excerpt of the stimulation trace is shown above the middle LFP trace). Bottom: LFP trace from the same rat before (left), during (middle), and after (right) test stimulation (1 min, 6 Hz, 2 mA), administered following a 30 min stimulation (6 Hz, 2 mA). (**b**) The somatosensory-evoked potential (SEP) of test stimulation at baseline (0 min, BL) and after 30 and 60 minutes (blue, red, and black, respectively, each averaged over 360 stimuli). (**c**) Somatosensory-evoked potential (SEP) in response to test stimulation at baseline (blue) and following a 30 min stimulation (red). (**d**) SEP in response to test stimulation at baseline (blue) and three time points following a 30 min stimulation (60 min, red; 90 min, green; 120 min, purple). (**e**) Maximum amplitude of the SEP (absolute values) following a 30 min stimulation compared to baseline (n = 6 rats, mean ± SEM, Wilcoxon, p=0.0312). (**f**) Area under the curve (AUC) of the SEP following a 30 min stimulation compared to baseline (n = 6 rats, mean ± SEM, Wilcoxon, p=0.0312). (**g**) Top: 1 hr LFP trace from a representative rat. Bottom: 5 s magnifications of the above trace at selected time points (noted by asterisks). Left to right: baseline activity; during test stimulation; following cortical application of 0.1 mM albumin (Alb);

*Figure 2 continued on next page*

*Figure 2 continued*

during test stimulation post-Alb. (**h**) SEP amplitude during test stimulation at baseline (normal aCSF, blue) and following 0.1 mM Alb (red). (**i**) Maximum amplitude of the SEP during test stimulation post-Alb compared to baseline (n = 5 rats, mean ± SEM, Wilcoxon, p=0.0312). (**j**) AUC of the SEP post-Alb compared to baseline. (n = 5 rats, mean ± SEM, Wilcoxon, p=0.0312). (**k**) Power spectrum density of 10 min spontaneous activity before (blue) and post-Alb (red) (p=0.0035, paired *t*-test). *p<0.05.

The online version of this article includes the following source data for figure 2:

**Source data 1.** All data measured and analyzed for *Figure 2*.

stimulation-induced transcytosis of albumin (*John et al., 2003*), we perfused the exposed cortex with methyl-β-cyclodextrin (mβCD), which inhibits albumin uptake by disrupting caveolae (*Schnitzer and Oh, 1994*; *Skotland et al., 2020*). Cortical application of mβCD prevented stimulus-induced BBB opening (*Figure 3c*), without affecting vascular response to stimulation (*Figure 3—figure supplement 2d*). mβCD also prevented stimulus-induced LTP (*Figure 3f–h*), further supporting the role of CMT-mediated albumin extravasation in activity-dependent plasticity in vivo.

## BBB modulation and LTP involve TGF-β signaling

Under pathological conditions, synaptogenesis and pathological plasticity that follow BBB dysfunction are mediated by the activation of TGF-βR1 signaling in astrocytes (*Weissberg et al., 2015*). TGF-β1 was also shown to directly induce BBB opening in endothelial cell cultures (*McMillin et al., 2015*) and regulate hippocampal synaptic plasticity (*Dahlmanns et al., 2023*; *Gradari et al., 2021*). We therefore tested the effect of stimulation in the presence of the specific TGF-βR1 blocker SJN2511 (SJN, 0.3 mM; *Gellibert et al., 2004*; *Weissberg et al., 2015*). Cortical exposure to SJN prevented stimulation-induced BBB opening (*Figure 3b and c*) with no effect on NVC (*Figure 3—figure supplement 2e*). SJN also prevented stimulation-induced LTP (*Figure 3f–h*). These results demonstrate that TGF-β signaling is required for activity-dependent BBB modulation, and the consequent activity-dependent plasticity, but not for NVC.

## Regulation of BBB transport and plasticity genes

To explore gene-expression changes related to our stimulation protocol, we conducted bulk RNA-sequencing transcriptome analysis of tissue dissected from the sensorimotor area of the cortex contralateral and ipsilateral to stimulated limb. Tissue harvested 1 and 24 hr post-stimulation (30 min) showed differentially expressed genes (DEGs) in the contralateral ('stimulated') compared to the ipsilateral ('non-stimulated') cortex (13.2 and 7.3%, 24 and 1 hr, respectively, *Figure 4a*, *Figure 4—figure supplement 1a*). To assess the variability between paired samples of stimulated and non-stimulated cortex of each animal at two time points following stimulation (24 hr vs. 1 hr), we used the Jensen–Shannon divergence metric (JSD) (*Sudmant et al., 2015*). JSD calculations using normalized counts indicated that the RNA expression 24 hr after stimulation was significantly different from that of rats 1 hr after stimulation (*Figure 4—figure supplement 1b*). Gene Ontology (GO) pathway enrichment analysis of DEGs from stimulated rats indicated that DEGs were primarily involved in synaptic plasticity processes (*Figure 4b*). A significantly higher number of DEGs related to synaptic plasticity was found in the contralateral cortex of stimulated rats compared to ipsilateral (p<0.001, *Figure 4d*). No significant differences were found in BBB or inflammation-related genes between the hemispheres, suggesting that the stimulation protocol is not associated with the transcriptional change observed in BBB dysfunction under pathological conditions (*Cacheaux et al., 2009*; *Kim et al., 2017*; *Figure 4d*). In contrast, a larger amount of transporter genes such as solute carrier transporters (Slc) and ATP-binding cassette (ABCs) families were differentially expressed in the stimulated compared to the non-stimulated cortex (*Figure 4—figure supplement 1c and d*). Both DEG and enrichment analysis were consistent with the activation of TGF-β signaling pathway (*Figure 4d*). Analysis of vascular cell-specific DEGs (*Vanlandewijck et al., 2018*) showed significantly more arterial smooth muscle cells-specific genes in the stimulated cortex (*Figure 4c*, *Figure 4—figure supplement 1e*). These results support the roles of stimulation-induced modulation of BBB transcellular transport and TGF-β signaling in activity-dependent plasticity (*Figure 5—figure supplement 1*).

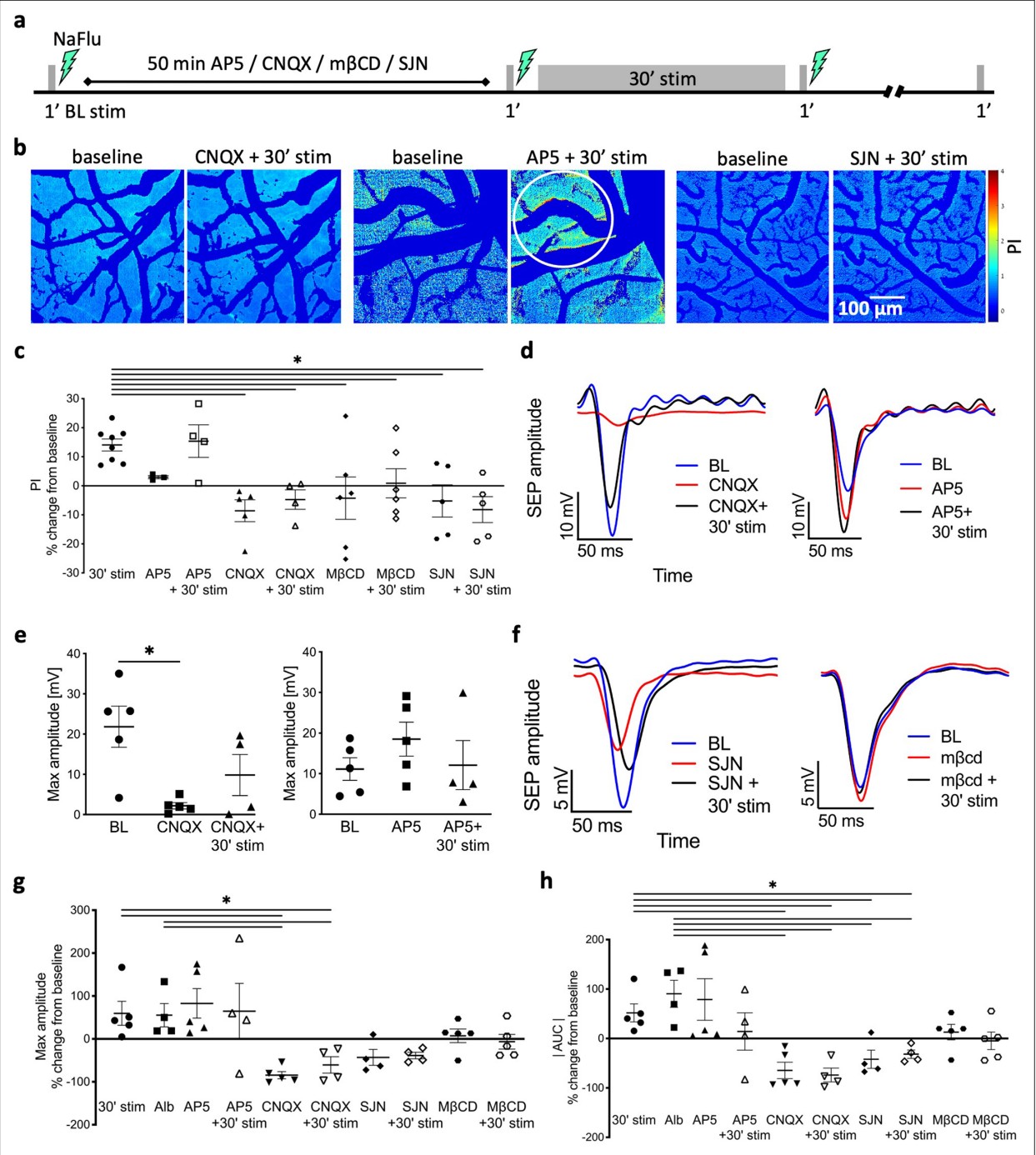

**Figure 3.** Stimulation-induced plasticity is associated with blood–brain barrier (BBB) modulation. (**a**) Timeline of the experimental protocol for stimulations and imaging with blockers application (CNQX/AP5 [50 µM]/mβCD [10 µM]/SJN [0.3 mM]). (**b**) NaFlu permeability maps of the cortical window before (control, left) and after CNQX/AP5/SJN + 30 min stimulation (right). Tracer accumulation area is marked with a white circle. (**c**) Permeability index (PI) following blockers application before and after stimulation compared to a 30 min stimulation. (% change from baseline, 30 min stim n = 8, AP5 n = 4, CNQX n = 5, mβCD n = 6, SJN n = 5; mean ± SEM, Kruskal–Wallis with false discovery rate [FDR] correction, *p<0.05). (**d**) Somatosensory-evoked potential (SEP) amplitude in response to test stimulation, baseline (blue); following application of CNQX (left, red) or AP5 (right, red); following CNQX + 30 min stimulation (left, black); and following AP5 + 30 min stimulation (right, black); (**e**) Left: max SEP amplitude to test stimulation following CNQX compared to baseline (mean ± SEM, n = 5, paired *t*-test, p=0.0176). Right: max SEP amplitude following AP5 compared to baseline. (**f**) SEP amplitude in response to test stimulation, baseline (blue); following SJN (red); following SJN + 30 min stimulation (black); and following mβCD (right). (**g**) Max SEP amplitude following 30 min stimulation or albumin application compared to blockers, and blockers + 30 min stimulation. (**h**) Area under the curve (AUC) of the SEP for test stimulation following 30 min stimulation or albumin application compared to blockers and blockers +

*Figure 3 continued on next page*

*Figure 3 continued*

30 min stimulation. (**g, h**) % change from baseline (mean ± SEM, 30 min stim n = 5, Alb n = 4, AP5 n = 5, CNQX n = 5, SJN n = 4, mβCD n = 5; Kruskal–Wallis with FDR correction, *p<0.05).

The online version of this article includes the following source data and figure supplement(s) for figure 3:

**Source data 1.** All data measured and analyzed for *Figure 3*.

**Figure supplement 1.** Modulation of blood–brain barrier (BBB) permeability associated with synaptic plasticity is activity dependent.

**Figure supplement 1—source data 1.** All data measured and analyzed for *Figure 3—figure supplement 1*.

**Figure supplement 2.** Stimulation-induced blood–brain barrier (BBB) modulation can be prevented with or without affecting the vascular response.

**Figure supplement 2—source data 1.** All data measured and analyzed for *Figure 3—figure supplement 2*.

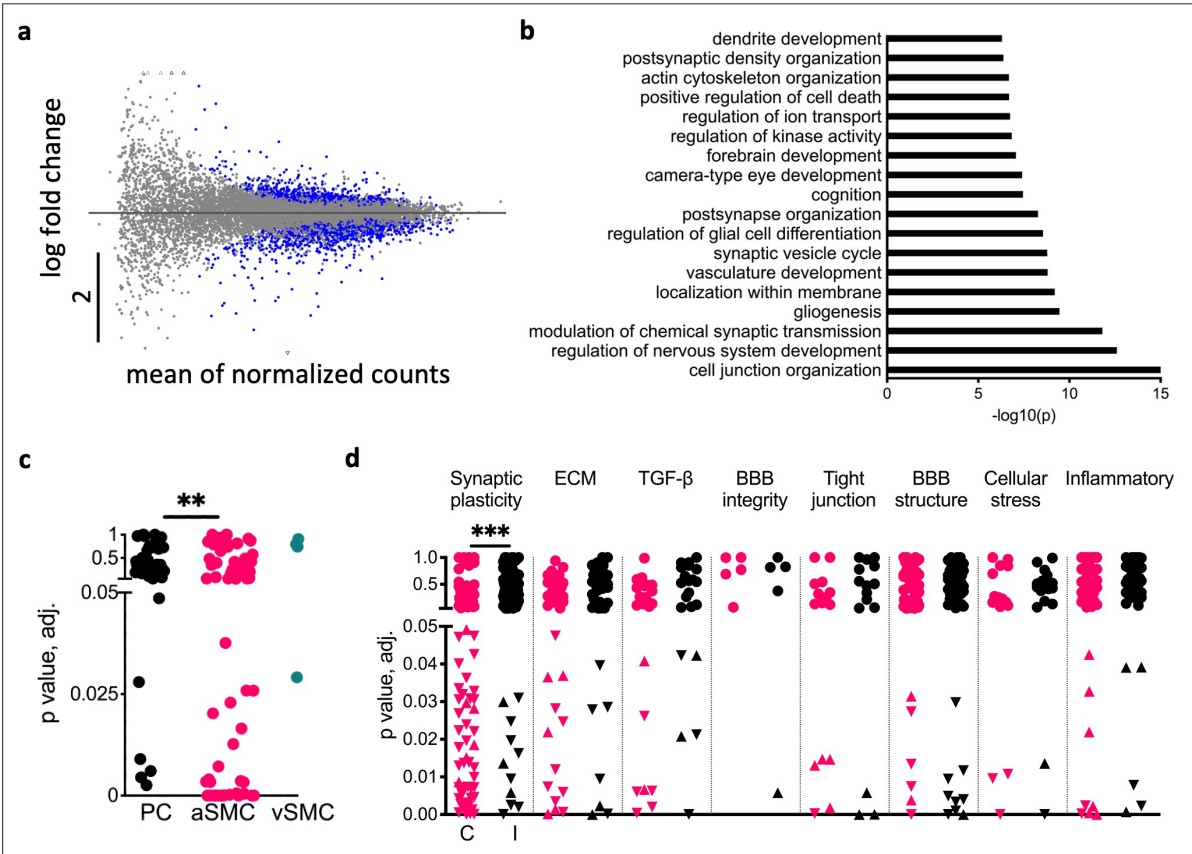

**Figure 4.** Neuronal activity regulates blood–brain barrier (BBB) transport and synaptic plasticity genes. (**a**) Scatter plot of gene expression from RNA-seq in the contralateral somatosensory cortex 24 vs. 1 hr after 30 min stimulation. The y-axis represents the log fold change, and the x-axis represents the mean expression levels (see 'RNA sequencing and bioinformatics'). Blue dots indicate statistically significant differentially expressed genes (DEGs) by Wald test (n = 8 rats per group). (**b**) Top Gene Ontologies (GO) enriched terms in the contralateral cortices of rats 24 vs. 1 hr after stimulation. (**c**) Vascular cell-specific DEGs to pericytes (PC), arterial smooth muscle cells (aSMC), and venous smooth muscle cells (vSMC) (PC n = 42, aSMC n = 70, p=0.0017, chi-square). (**d**) Scatter plot of DEGs divided by groups of interest: BBB properties, neurovascular unit (NVU) properties, Synaptic plasticity, and inflammatory-related genes in the contralateral (red) vs. ipsilateral (black) cortices of stimulated rats. Circles represent genes with no significant differences between 1 and 24 hr post-stimulation. Upward and downward triangles indicate significantly up- and downregulated genes, respectively. The contralateral somatosensory cortex was found to have a significantly higher number of DEGs related to synaptic plasticity than the ipsilateral side (***p<0.001, chi-square).

The online version of this article includes the following source data and figure supplement(s) for figure 4:

**Source data 1.** All data measured and analyzed for *Figure 4*.

**Figure supplement 1.** Neuronal activity regulates blood–brain barrier (BBB) transport and neurovascular unit (NVU) cells-specific genes.

**Figure supplement 1—source data 1.** All data measured and analyzed for *Figure 4—figure supplement 1*.

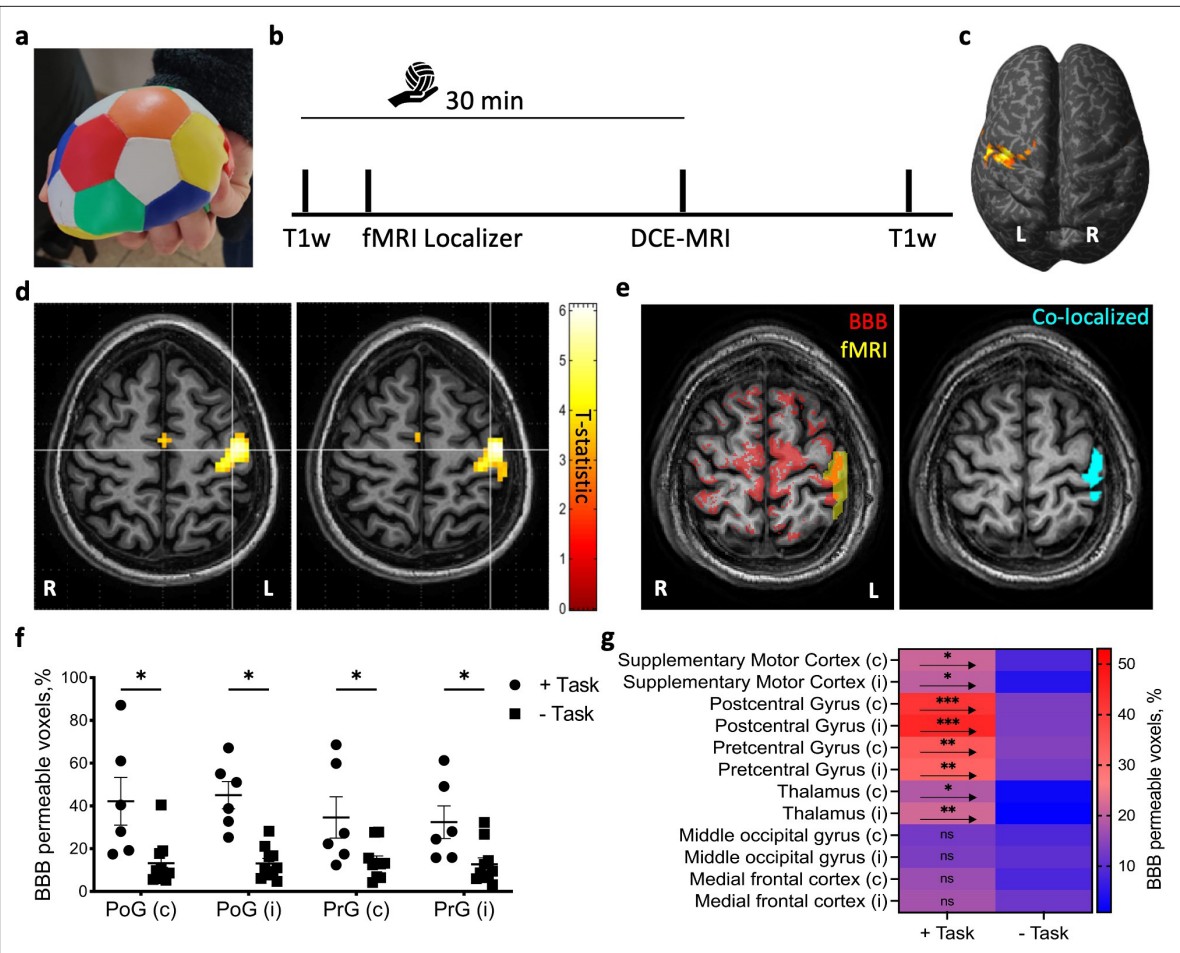

**Figure 5.** Cortical activation in fMRI co-localizes with blood–brain barrier (BBB) modulation. (**a**) Subjects were given an elastic stress ball to squeeze continuously for the length of the session (30 min). (**b**) Timeline of the experimental protocol for task performance (the hand holding a ball represents the duration of the ball-squeezing task), and MRI sequences (T1-weighted, functional [fMRI], and dynamic contrast-enhanced [DCE] MRI. (**c**) Activation map (voxels with statistically significant activation [*t*-statistic]) for the localizer task displayed over the inflated brain of an exemplary subject (p<0.05, family-wise error [FWE] corrected, neurological convention). (**d**) Activation maps for the localizer task displayed over anatomical axial slices of an exemplary subject (p<0.05, FWE corrected, radiological convention). White lines point to voxels of highest activation (*t*-statistic). (**e**) Left: superimposed masks of BBB modulated voxels (red) and fMRI activation (yellow). Right: co-localized voxels map on the same slice (cyan). (**f**) The percent of voxels with BBB leakage in the primary motor cortex (M1, precentral gyrus, PrG) and primary somatosensory cortex (S1, postcentral gyrus, PoG) for subjects performing the task compared to controls (Task n = 6, controls n = 10, i, ipsi, c, contra, mean ± SEM, *p<0.05, two-way ANOVA with false discovery rate [FDR] correction). (**g**) Heatmap of BBB modulated voxels percentage (as in **f**) in motor/sensory-related areas and some non-activated cortical regions of task vs. controls (+Task n = 6, -Task n = 10, i, ipsi, c, contra, mean ± SEM, *p<0.05, **p<0.001, ***p<0.0001, ns, nonsignificant, two-way ANOVA with FDR correction).

The online version of this article includes the following source data and figure supplement(s) for figure 5:

**Source data 1.** All data measured and analyzed for *Figure 5*.

**Figure supplement 1.** Activity-dependent modulation of blood–brain barrier (BBB) permeability associated with synaptic plasticity.

## Physiological BBB modulation in the human brain

Finally, we tested for neuronal activity-induced BBB modulation in healthy human volunteers by measuring the blood-oxygenation-level-dependent (BOLD) response to a 30 min stress ball squeeze task and blood-to-brain transport using dynamic contrast-enhanced (DCE) MRI (*Figure 5a and b*). As expected, BOLD response was found in the pre- and post-central gyrus, corresponding with the primary motor and sensory cortices, respectively, of the hemisphere contralateral to the squeezing hand (*Figure 5c and d*). DCE-MRI showed higher BBB permeability to the gadolinium-based contrast agent gadoterate meglumine (Dotarem) in sensory/motor-related areas of subjects performing the task compared to controls (*Figure 5e–g*). Non-activated cortical regions such as the medial frontal

cortex and middle occipital gyrus did not differ in BBB permeability between task and controls (*Figure 5g*). A comparison of the spatial distribution of functional and permeability changes revealed a higher percentage of activated regions with modulated BBB in task performers than in control subjects (*Figure 5e–g*).

## Discussion

This study demonstrates that prolonged physiological stimulation modulates BBB permeability in the rat and human cortex. In rats, we show that prolonged paw stimulation results in increased cross-BBB extravasation of both low- and high-molecular-weight tracers. We also show that this extravasation is (1) focal and localized to the cortical area corresponding to the stimulation; (2) mediated by AMPA neurotransmission, TGF-β signaling, and caveolae transcytosis; and (3) associated with synaptic potentiation and increased postsynaptic density. In humans, we present the first evidence for increased BBB permeability in response to prolonged limb activity. Together, our results suggest that stimulation-induced BBB modulation may play a role in synaptic potentiation.

Physiological changes in cross-BBB transport have been previously reported in response to whisker stimulation, circadian rhythms, and age (*Nishijima et al., 2010*; *Pulido et al., 2020*; *Yang et al., 2020*). Here, we examined whether the mechanisms involved in this process may include those identified in our studies of pathological BBB modulation. Previous studies from our group showed that brain injury is associated with albumin transcytosis, albumin binding to TGF-βR in astrocytes, and pathological TGF-β-mediated plasticity (*Salar et al., 2016*). Here, we show that prolonged physiological sensorimotor activation is also associated with albumin extravasation (although at a significantly lower concentration than under pathological conditions; *Figure 1l*, *Figure 1—figure supplement 1d*) and TGF-βR signaling. We further show that blocking caveolae-mediated albumin transcytosis or TGF-βR signaling inhibits stimulation-evoked increase in BBB permeability and synaptic potentiation (without affecting the physiological hemodynamic response). These results suggest that caveolae albumin transcytosis and TGF-β signaling are key mediators of activity-dependent BBB modulation and subsequent synaptic plasticity. These findings are supported by reports linking TGF-β signaling to synaptic plasticity induced by environmental enrichment *Dahlmanns et al., 2023*; enhanced spine and synapse formation *Patel and Weaver, 2021*; and hippocampal neuroplasticity (*Gradari et al., 2021*). Moreover, our finding of caveolae-mediated albumin transcytosis in pial arterioles is in line with previous studies showing caveolae-dependent transcytosis of albumin *Tecedor et al., 2013*; caveolae abundance in arteriolar ECs and not capillary ECs *Chow et al., 2020*; and the contribution of caveolae transcytosis to NVC (*Andreone et al., 2017*; *Chow et al., 2020*). Notably, in our experiments, caveolae inhibition prevented BBB modulation yet did not significantly affect NVC, likely due to the contribution of other vasodilatory pathways such as endothelial nitric oxide synthase.

As mentioned above, albumin is a known activator of TGF-β signaling, and TGF-β has a well-established role in neuroplasticity. Interestingly, emerging evidence suggests that TGF-β also increases cross-BBB transcytosis (*Betterton et al., 2022*; *Kaplan et al., 2020*; *McMillin et al., 2015*; *Schumacher et al., 2023*). Hence, we propose the following two-part hypothesis for the TGF-β/BBB-mediated synaptic potentiation observed in our experiments: (1) prolonged stimulation triggers TGF-β signaling and increased CMT of albumin; and (2) extravasated albumin induces further TGF-β signaling, leading to synaptogenesis and additional cross-BBB transport in a self-reinforcing positive feedback loop. Future research is needed to examine the validity of this hypothesis.

Our study also shows that physiological BBB modulation requires AMPAR glutamate signaling and not NMDA. This finding is consistent with studies showing the dominant role of AMPAR in electrophysiological and hemodynamic responses to somatosensory stimulation (*Graves et al., 2021*; *Gsell et al., 2006*; *Zhang et al., 2015*). Moreover, the well-established role of AMPAR in neuroplasticity, LTP, and experience-dependent sensory cortical plasticity (*Daw et al., 1993*; *Feldman, 2009*; *Watt et al., 2004*) further supports the plasticity-related role of activity-dependent BBB modulation.

Our gene analysis was designed to complement our in vivo and histological findings by assessing the magnitude of change in DEGs. This analysis showed that (1) the hemisphere contralateral to the stimulus has significantly more DEGs than the ipsilateral hemisphere; and (2) the DEGs were related to synaptic plasticity and TGF-β signaling. These findings strengthen the hypothesis raised by our experiments.

Our animal experiments show that a 30 min limb stimulation (at 6 Hz and 2 mA) increases cross-BBB influx, while a 1 min stimulation (of similar frequency and magnitude) does not. We argue that both types of stimulations fall within the physiological range because in rats activity between 5 and 15 Hz represents physiological rhythmic whisker movement during environment exploration (*Mégevand et al., 2009*). Moreover, our continuous electrophysiological recordings showed no signs of epileptiform or otherwise pathological activity and the recorded SEP levels were similar to those reported in previous LTP studies in rats (*Eckert and Abraham, 2010*; *Han et al., 2015*; *Mégevand et al., 2009*) and humans (*McGregor et al., 2016*). In humans, skill acquisition often involves motor training sessions that last ≥30 min (*Bengtsson et al., 2005*; *Classen et al., 1998*) and result in physiological plasticity of sensory and motor systems (*Classen et al., 1998*; *Draganski et al., 2004*; *Sagi et al., 2012*). Hence, the experimental task in our human study (30 min of repetitive squeezing of an elastic stress ball) is likely to represent physiological learning task involving neuronal activation in primarily motor and sensory areas (*Halder et al., 2005*). Future human and animal studies are needed to explore the BBB modulating effects of additional stimulation protocols, with varying durations, frequencies, and magnitudes. Such studies may also elucidate the temporal and ultrastructural characteristics that differentiate between physiological and pathological BBB modulation.

A key limitation of our animal experiments is the fact they were performed under anesthesia due to the complex nature of the experimental setup (i.e., simultaneous cortical imaging and electrophysiological recordings). Anesthetic agents can potentially alter neuronal activity, SEPs, CBF, and vascular responses (*Aksenov et al., 2015*; *Lindauer et al., 1993*; *Masamoto and Kanno, 2012*). To minimize these effects, we used ketamine-xylazine anesthesia, which, unlike other anesthetics, was shown to maintain robust BOLD and SEP responses to neuronal activation (*Franceschini et al., 2010*; *Shim et al., 2018*). Notably, the antagonistic effect of ketamine on NMDA receptors might attenuate the magnitude of SEP potentiation recorded in our experiments (*Anis et al., 1983*; *Salt et al., 1988*).

Another limitation of our animal study is the potentially non-specific effect of mβCD – an agent that disrupts caveola transport but may also lead to cholesterol depletion (*Keller and Simons, 1998*). To mitigate this issue, we used a very low mβCD concentration (10 µM), orders of magnitude below the concentration reported to deplete cholesterol (*Koudinov and Koudinova, 2001*). Lastly, our animal study is limited by the inclusion of solely male rats. While our findings in humans did not point to sex-related differences in stimulation-evoked BBB modulation, larger animals and human studies are needed to examine this question.

To conclude, our study suggests that BBB modulation in response to physiological neuronal activity involves mechanisms previously identified in pathology-induced BBB changes (despite the significantly smaller increase in cross-BBB influx under physiological conditions). Our findings further suggest that physiological modulation of cross-BBB influx may play a role in synaptic plasticity, calling for future research into the dynamics of BBB function in health and disease.

## Materials and methods
### Study design

This study included rat and human experiments. The rat experiments were designed to explore the mechanisms involved in altered BBB permeability following physiological stimulation and the potential link to synaptic plasticity. The experimental paradigm involved a 30 min rat limb stimulation at a frequency comparable to rhythmic whisker activity during environment exploration (*Mégevand et al., 2009*). The effects of stimulation on BBB permeability were studied using in vivo and histopathological imaging. The effects of stimulation on synaptic plasticity were evaluated by examining the SEPs in response to a 1 min test stimulation before and after the 30 min stimulation. A subset of experiments involved the application of blockers to test their effect on BBB permeability and synaptic plasticity. Another subset of experiments was designed to test the effects of stimulation on gene expression.

In humans, the experimental paradigm involved a 30 min session of squeezing a stress ball. fMRI was used to localize the brain area activated during this motor task, and DCE MRI was used to evaluate BBB permeability post stimulation.

All experiments and data analysis were performed blindly to the treatment group.

## Animal preparation and surgical procedures

All experimental procedures were performed according to approved Institutional Animal Care and Use Committee (IACUC) protocol IL-50-07-2020 of the Ben-Gurion University of the Negev, Beer-Sheva, Israel.

Adult Sprague–Dawley male rats (300–350 g; Harlan Laboratories) were kept under a 12:12 hr light and dark regimen and supplied with drinking water and food ad libitum. Surgical procedures were performed as reported previously (*Prager et al., 2010*). Rats were deeply anesthetized by intraperitoneal administration of ketamine (100 mg/ml, 0.08 ml/100 g) and xylazine (20 mg/ml, 0.06 ml/100 g). The tail vein was cannulated, and animals were placed in a stereotactic frame under a SteREO Lumar V12 fluorescence microscope (Zeiss). Body temperature was continuously monitored and kept stable at 37 ± 0.5°C using a feedback-controlled heating pad (Physitemp). Heart rate, breath rate, and oxygen saturation levels were continuously monitored using MouseOx (STARR Life Sciences). A cranial section (4 mm caudal, 2 mm frontal, 5 mm lateral to bregma) was removed over the right sensorimotor cortex. The dura and arachnoid layers were removed, and the exposed cortex was continuously perfused with artificial CSF (aCSF [*Prager et al., 2010*], containing [in mM]: 124 NaCl, 26 NaHCO$_3$, 1.25 NaH$_2$PO$_4$, 2 MgSO$_4$, 2 CaCl$_2$, 3 KCl, and 10 glucose, pH 7.4).

## Stimulation protocol

The left forelimb or hindlimb of the rat was stimulated using Isolated Stimulator device (AD Instruments) attached with two subdermal needle electrodes (0.1 ms square pulses, 2–3 mA) at 6 Hz frequency. Test stimulation consisted of 360 pulses (60 s) and delivered before (as baseline) and after long-duration stimulation (30 min, referred throughout the text as 'stimulation'). In control and albumin rats, only short-duration stimulations were performed. Under sham stimulation, electrodes were placed without delivering current.

## Electrophysiology

### In vivo recordings

Stimulus-evoked potentials were recorded in the somatosensory cortex. LFP and SEP were recorded using a glass microelectrode (1.5 × 1.1 mm, 1–2 μm tip, 2–10 MΩ) filled with aCSF, which was inserted with an isolated, chlorinated silver wire connected to a headstage and amplifier (EXT-02B, npi electronic, Germany). The electrode was attached to a digital micromanipulator (MP-225, Sutter Instruments, CA) and carefully placed over the identified area of the sensorimotor response to the stimulation.

### Analysis of electrophysiological recordings

Data acquisition was performed using PowerLab and LabChart (AD Instruments), for stimulation, and extracellular recordings. Signals were digitized (1 kHz) and filtered (high-pass 1 Hz, low-pass 45 or 300 Hz). Analyses were performed using in-house MATLAB scripts. Synaptic plasticity was then quantified using the following two measures: maximal amplitude of the dominant negative peak and the absolute AUC of the SEP waveform (i.e., all positive and negative peaks between 10 and 160 ms post-stimulus period). To overcome potential variability in SEP morphology between animals (*Mégevand et al., 2009*), each animal's plasticity measures (max amplitude and AUC of post-stimulation SEP) were compared to the same measures at baseline.

For power analysis in albumin experiments, 10 min segments recorded before starting and after terminating albumin perfusion were used.

## Drug application

In some experiments, the following blockers were added to the aCSF and continuously perfused to the cortical window for 50 min: D-(-)–2-amino-5-phosphonopentanoic acid (AP5, 50 μM), 6-cyano-7-nitroquinoxaline-2,3-dione (CNQX, 50 μM), methyl-β-cyclodextrin (MβCD, 10 μM), or SJN2511 (SJN, 0.3 mM [*Weissberg et al., 2015*]). To simulate BBB opening, bovine serum albumin (Alb, 0.1 mM [*Seiffert et al., 2004*]) was added to the aCSF and continuously perfused to the cortical window for 30 min. Experimenter was blind to the selected compound.

## In vivo imaging

Dynamic imaging of regional cerebral blood flow, optical imaging of cortical oxygenation, NVC, and vessel diameter measurements were carried out as previously described (*Bouchard et al., 2009*; *Levi et al., 2012*; *Prager et al., 2010*). Before, during, and after electrical limb stimulation, full-resolution (2560 × 2,160 px) images were obtained for regional localization of the hemodynamic response, followed by acquisition of localized (512 × 512 px) images of cortical surface vessels (2 frames/s, EMCCD camera, Neo DC-152Q-COO-FI; Andor Technology) under 525/50 band pass filter (38 HE, Zeiss). Images were analyzed offline using in-house MATLAB (MathWorks) scripts.

Fluorescent angiography and BBB permeability measurements and analyses were performed as described previously (*Prager et al., 2010*; *Vazana et al., 2016*). The non-BBB permeable fluorescent dye sodium fluorescein (NaFlu) was injected intravenously (0.2 ml, 1 mg/ml). Images of cortical surface vessels were obtained before, during, and after tracer injection at 5 frames/s. Across all experiments, acquired images were the same size (512 × 512 pixel, ~1 × 1 mm), centered above the responding arteriole. Images were analyzed offline using MATLAB as described (*Vazana et al., 2016*). Briefly, image registration and segmentation were performed to produce a binary image, separating blood vessels from extravascular regions. For each extravascular pixel, a time curve of signal intensity over time was constructed. To determine whether an extravascular pixel had tracer accumulation over time (due to BBB permeability), the pixel's intensity curve was divided by that of the responding artery (i.e., the arterial input function, representing tracer input). This ratio was termed the BBB permeability index (PI), and extravascular pixels with PI > 1 were identified as pixels with tracer accumulation due to BBB permeability.

## Histology and immunohistochemistry

Albumin extravasation was confirmed histologically in separate cohorts of rats that were anesthetized and stimulated without craniotomy surgery. Assessment of albumin extravasation was performed using a well-established approach that involves peripheral injection of either labeled albumin (bovine serum albumin conjugated to Alexa Flour 488, Alexa488-Alb) or albumin-labeling dye (Evans blue, EB – a dye that binds to endogenous albumin and forms a fluorescent complex), followed by histological analysis of brain tissue (*Ahishali and Kaya, 2020*; *Ivens et al., 2007*; *Lapilover et al., 2012*; *Obermeier et al., 2013*; *Veksler et al., 2020*). Since extravasated albumin is taken up by astrocytes (*Ivens et al., 2007*; *Obermeier et al., 2013*, and others), it can be visualized in the brain neuropil after brain removal and fixation (*Ahishali and Kaya, 2020*; *Ivens et al., 2007*; *Lapilover et al., 2012*; *Veksler et al., 2020*). Five rats were injected with Alexa488-Alb (1.7 mg/ml) and five with EB (2%, 20 mg/ml, n = 5). The injections were administered via the tail vein. Following injection, rats were transcardially perfused with cold phosphate-buffered saline (PBS), followed by PBS containing 4% paraformaldehyde (PFA). Brains were then removed, fixed overnight (4% PFA, 4°C), cryoprotected with sucrose gradient (10% followed by 20 and 30% sucrose in PBS), and frozen in optimal cutting temperature compound. Coronal sections (40 µm thick) were obtained using a freezing microtome (Leica Biosystems) and imaged for dye extravasation using a fluorescence microscope (Axioskop 2; Zeiss) equipped with a CCD digital camera (AxioCam MRc 5; Zeiss).

For immunostaining, brain slices were incubated in blocking solution (5% donkey serum in 0.1% Triton X-100/Tris-buffered saline [TBS]), then incubated in primary antibody, followed by secondary antibody. Finally, slices were mounted and stained for DAPI (DAPI-Fluoromount-G, Invitrogen) to label nuclei. Staining was performed against glial fibrillary acidic protein (mouse anti GFAP–Cy3, Sigma-Aldrich, C9205), Secondary antibody was anti-mouse Alexa Fluor 488 (Thermo Fisher, A21206).

## Immunoassays
### ELISA for albumin extravasation

To assess the degree of albumin extravasation following stimulation, 40 adult Sprague–Dawley male rats (200–300 g) were randomly divided into five groups. Rats were anesthetized (ketamine [75 mg/kg] and xylazine [5 mg/kg]) and underwent either a 30 min stimulation, sham stimulation (electrodes placed without current delivery), or photothrombosis stroke (PT) as previously described (*Lippmann et al., 2017*; *Schoknecht et al., 2014*). Briefly, rose bengal was administered intravenously (20 mg/kg) and a halogen light beam was directed for 15 min onto the intact exposed skull over the right somatosensory cortex. Then, rats were deeply anaesthetized and transcardially perfused with cold PBS, and

brains were extracted at three time points following stimulation (immediately [real and sham], 4 and 24 hr post stimulation) and 24 hr post stroke. The ipsi- and contralateral somatosensory cortices were dissected, weighed, and snap-frozen on dry ice. Fresh frozen samples were then homogenized in PBS + 1% Triton X-100 (v/v). Tissue homogenates were centrifuged at 13,000 × $g$ for 10 min and supernatants were stored at –80°C until further analysis. Albumin levels were determined in diluted samples using the Rat Albumin ELISA kit (Bethyl Laboratories, TX) according to the manufacturer's instructions.

## Western blotting

Brain tissues were collected from stimulated and control rats as described above. Protein lysates were separated using 10% sodium dodecyl sulfate polyacrylamide gel electrophoresis (Bio-Rad) and then transferred to nitrocellulose membranes (Bio-Rad). Membranes were washed with TBS + 0.1% Tween20 (TBST, 3 × 5 min), then blocked with 5% BSA in TBS for 1 hr. Membranes were then incubated overnight at 4°C with primary antibodies, followed by 2 hr incubation with secondary antibodies. Protein bands were visualized with the enhanced chemiluminescent HRP substrate Crescendo Immobilon (Millipore). Analysis was performed in ImageJ, and band density levels were expressed as fold changes to controls after normalization to β-actin and controls within the same blot. Primary antibodies used were mouse anti-PSD-95 (1:1500, Thermo Fisher Scientific, MA1-045) and anti-β-actin (loading control, 1:2000, Abcam, ab8226). Secondary antibodies used were goat anti-mouse (1:4000, Abcam, ab205719) and goat anti-rabbit (1:4000, Abcam, ab97051).

## RNA extraction and purification

Total RNA was extracted from 100 to 150 mg of flash-frozen rat somatosensory cortex tissue using TRIzol reagent (Thermo Fisher Scientific #15596026) as previously described (*Rio et al., 2010*). Glyco-blue (Invitrogen #AM9515) was added to assist with RNA pellet isolation. Pellets were washed with 75% ethanol and allowed to dry before being dissolved in 20 µl of RNase-DNase free water. 2 µl of 10× DNase I Buffer (New England Biolabs, #m0303s), 1 µl of rDNase I (Thermo Fisher Scientific, AM2235), and DNase Inactivation Reagent (Thermo Fisher Scientific, AM1907) were added to each RNA sample for purification.

## RNA sequencing and bioinformatics

Library preparation was performed by the QB3-Berkeley Functional Genomics Laboratory at UC Berkeley. mRNA selection was done with the oligo-dT beads. Both total RNA and oligo dT bead-selected mRNA quality were assessed on an Agilent 2100 Bioanalyzer. Libraries were prepared using the KAPA RNA Hyper Prep kit (Roche KK8540). Truncated universal stub adapters were ligated to cDNA fragments, which were then extended via PCR using unique dual-indexing primers into full-length Illumina adapters. Library quality was checked on an AATI (now Agilent) Fragment Analyzer. Libraries were then transferred to the QB3-Berkeley Vincent J. Coates Genomics Sequencing Laboratory, also at UC Berkeley. Library molarity was measured via quantitative PCR with the KAPA Library Quantification Kit (Roche KK4824) on a Bio-Rad CFX Connect thermal cycler. Libraries were then pooled by molarity and sequenced on an Illumina NovaSeq 6000 S4 flowcell for 2 × 150 cycles, targeting at least 25M reads per sample. Fastq files were generated and demultiplexed using Illumina bcl2fastq2 v2.20. No other processing, trimming, or filtering was conducted. RNA-seq reads were aligned using STAR v2.9.1a against UCSC rat genome rn6. Downstream analysis was performed using FastQC v0.11.9, MultiQC v1.11, and featureCounts. RStudio was used for data analysis, R v4.0.2. with the packages tidyverse v1.3.1, IHW, DESeq2. GO analysis of enriched genes in ipsi- and contralateral somatosensory cortices of stimulated rats was performed using Metascape (*Zhou et al., 2019*; http://metascape.org/). Analysis of cell-specific and vascular zonation genes was performed as described (*Vanlandewijck et al., 2018*) using the database provided in http://betsholtzlab.org/VascularSingleCells/database.html.

## Human study

### Participants

Male and female healthy individuals, aged 18–35, with no known neurological or psychiatric disorders were recruited to undergo MRI scanning while performing a motor task (n = 6; three males and three females). MRI scans of 10 sex- and age-matched individuals (with no known neurological or

psychiatric disorders) who did not perform the task were used as control data (n = 10; five males and five females).

All procedures were approved by the Soroka University Medical Center Institutional Review Board (0121-17-SOR). All participants gave their written informed consent before participation.

## MRI

MRI scans were performed in the imaging center at the Soroka Medical Center using a 3T Philips Ingenia MRI scanner. A high-resolution T1-weighted anatomical scan (3D gradient echo, TE/TR 3.7/8.2 ms, voxel size of 1 mm³, 256 × 256 acquisition matrix), followed by a T2-weighted scan (TE/TR = 90/3000 ms, voxel size 0.45 × 0.45 × 4 mm). Functional data were collected using a gradient echo EPI, with voxel size of 3 × 3 × 3 mm, TE/TR = 35/2000 ms, 83 × 80 acquisition matrix, flip angle of 90°, and FOV 109 mm. For calculation of pre-contrast longitudinal relaxation times, variable flip angle method was used (3D T1w-FFE, TE/TR = 2/10 ms, acquisition matrix: 256 × 256, voxel size: 0.89 × 0.89 × 6 mm, flip angles: 5, 15, 20, and 25°). DCE sequence was then acquired (Axial, 3D T1w-FFE, TE/TR = 2/4 ms, acquisition matrix: 192 × 187 [reconstructed to 256 × 256], voxel size: 0.9 × 0.9 × 6 mm, flip angle: 20°, $\Delta t$ = 10 s, temporal repetitions: 100, total scan length: 16.7 min). An intravenous bolus injection (0.1 mmol/kg, 0.5 M) of the gadolinium-based contrast agent gadoterate meglumine (Dotarem, Guerbet, France) was administered using an automatic injector after five dynamic scans at a rate of 1.5 ml/s.

### fMRI localizer motor task

fMRI was used to localize the brain area activated in response to the experimental motor task. During the fMRI scan, participants were asked to squeeze the stress ball on cue (with green screen indicating Go and red screen indicating Stop) for a total of five experimental blocks. Task-related activation was evaluated for each cue, and voxels with statistically significant activation (t-statistic) were used to construct an 'activation map'. The statistical threshold for clusters was set at p<0.05 and corrected for multiple comparisons using family-wise error (FWE) over the whole brain (uncorrected threshold p<0.001) to minimize detection of false positives (type I error).

### Preprocessing of functional data

fMRI data preprocessing was performed using the Statistical Parametric Mapping package (SPM12; Wellcome Trust Centre for Neuroimaging, London, UK) for MATLAB. The first 10 functional images of each run series were discarded to allow stabilization of the magnet. Functional images were realigned and co-registered to the T1-weighted structural image, followed by segmentation and normalization to the Montreal Neurological Institute (MNI) space. Spatial smoothing using Gaussian kernel (FWHM of 6 mm) was performed.

### BBB permeability quantification

Analysis was performed as reported (*Serlin et al., 2019*; *Veksler et al., 2020*). Briefly, preprocessing included image registration and normalization to MNI space using SPM12. BBB permeability was calculated using in-house MATLAB script. A linear fit was applied to the later stage of the scan and the slope of contrast agent concentration changes over time was calculated for each voxel. Positive slopes reflected contrast agent accumulation due to BBB modulation. To compensate for physiological (e.g., heart rate, blood flow) and technical (e.g., injection rate) variability, slopes were normalized to the slope of each subject's transverse sinus. Values were considered to reflect modulated BBB when exceeding the 95th percentile of the corresponding mean cumulative distribution function from healthy young controls. Voxels that reflect modulated BBB values were color-coded and superimposed on the anatomical scans to illustrate the kinetics, localization, and extent of BBB pathology.

## Statistical analysis

All statistical tests were performed using Prism9 (GraphPad Software) and are indicated in the figure legends. Tests were two-tailed and corrected for multiple comparisons with two-stage linear step-up false discovery rate (FDR) procedure of Benjamini, Krieger, and Yekutieli whenever applicable. Paired tests were used for inter-hemispheric comparisons. Nonparametric tests were used for data which was not normally distributed. Differences in fluorescence in *Figure 1j and k* were tested using a nested

*t*-test. Differences in BBB PI, AUC, and maximum amplitude of the SEP before and after stimulation were tested using the Wilcoxon matched-pairs signed-rank test. Differences between groups in ELISA, AUC, and maximum amplitude of the SEP were analyzed using Kruskal–Wallis test with FDR correction. Differences in the amount of DEGs in *Figure 4c and d* and *Figure 4—figure supplement 1c–e* were analyzed using chi-square contingency test. Differences in BBB permeability across regions in the human brain were analyzed using two-way ANOVA with FDR correction. p≤0.05 was defined as the statistical significance level. Analyses were blind to experimental conditions.

## Acknowledgements

This study was supported by the Israel Science Foundation (Awards Nos. 717/15 and 2254/20), Canadian Institutes of Health Research (CIHR Award No. PJT 148896), and the European Research Area Network Neuron Award (Award No. NDD 168164). We thank QB3 Genomics Sequencing Laboratory, UC Berkeley, Berkeley, CA, RRID:SCR_022170 for using their facilities.

## Additional information

### Funding

| Funder | Grant reference number | Author |
| --- | --- | --- |
| Israel Science Foundation | 717/15 | Alon Friedman |
| Israel Science Foundation | 2254/20 | Alon Friedman |
| Canadian Institutes of Health Research | PJT 148896 | Alon Friedman |
| European Research Area Network | NDD 168164 | Alon Friedman |

The funders had no role in study design, data collection and interpretation, or the decision to submit the work for publication.

### Author contributions

Evyatar Swissa, Conceptualization, Data curation, Software, Formal analysis, Validation, Investigation, Visualization, Methodology, Writing - original draft, Project administration, Writing - review and editing; Uri Monsonego, Formal analysis, Investigation, Visualization; Lynn T Yang, Data curation, Software, Formal analysis, Investigation, Visualization, Writing - review and editing; Lior Schori, Itamar Burger, Investigation; Lyna Kamintsky, Software, Formal analysis, Validation, Visualization, Writing - review and editing; Sheida Mirloo, Rishi Patel, Formal analysis; Sarit Uzzan, Investigation, Methodology; Peter H Sudmant, Formal analysis, Visualization, Methodology; Ofer Prager, Conceptualization, Resources, Supervision, Validation, Investigation, Methodology, Project administration, Writing - review and editing; Daniela Kaufer, Resources, Supervision, Funding acquisition, Methodology, Project administration, Writing - review and editing; Alon Friedman, Conceptualization, Resources, Supervision, Funding acquisition, Methodology, Writing - original draft, Project administration, Writing - review and editing

### Author ORCIDs

Evyatar Swissa ⓘ http://orcid.org/0000-0002-6644-7230
Peter H Sudmant ⓘ http://orcid.org/0000-0002-9573-8248
Daniela Kaufer ⓘ http://orcid.org/0000-0002-3830-5999
Alon Friedman ⓘ http://orcid.org/0000-0003-4780-8456

### Ethics

All procedures were approved by the Soroka University Medical Center Institutional Review Board (0121-17-SOR). All participants gave their written informed consent before participation.
All experimental procedures were performed according to approved Institutional Animal Care and Use Committee (IACUC) protocol IL-50-07-2020 of the Ben-Gurion University of the Negev, Beer-Sheva, Israel.

Reviewer #3 (Public Review): https://doi.org/10.7554/eLife.89611.4.sa1
Author response https://doi.org/10.7554/eLife.89611.4.sa2

## Additional files

### Supplementary files
• MDAR checklist

### Data availability
Source data files for quantification described in the text or shown in figures and figure supplements are provided in the manuscript. Sequencing data have been deposited in GEO under accession code GSE223282.

The following dataset was generated:

| Author(s) | Year | Dataset title | Dataset URL | Database and Identifier |
|---|---|---|---|---|
| Yang LT, Swissa E, Kaufer D, Friedman A | 2024 | Use-dependent synaptic plasticity associates with modulation of blood-brain-barrier permeability in the somatosensory cortex | https://www.ncbi.nlm.nih.gov/geo/query/acc.cgi?acc=GSE223282 | NCBI Gene Expression Omnibus, GSE223282 |

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
