## [Editor Report · eLife assessment]

This study builds upon previous work which demonstrated that brain injury results in the entry of a protein called albumin into the brain which then causes diverse effects. The present study shows that prolonged stimulation of a forelimb in a rat leads to albumin entry and is associated with effects that suggest plasticity is enhanced in the stimulated side of the brain. The strength of evidence was **convincing** and results are **important** because they suggest a previously considered pathological process may be relevant to the normal brain and have benefits.

---

## [Referee Report · Reviewer #3 (Public Review)]

Summary:

This study used prolonged stimulation of a limb to examine possible plasticity in somatosensory evoked potentials induced by the stimulation. They also studied the extent that the blood brain barrier (BBB) was opened by the prolonged stimulation and whether that played a role in the plasticity. They found that there was potentiation of the amplitude and area under the curve of the evoked potential after prolonged stimulation and this was long-lasting (>5 hrs). They also implicated extravasation of serum albumin, caveolae-mediated transcytosis, and TGFb signalling, as well as neuronal activity and upregulation of PSD95. Transcriptomics was done and implicated plasticity related genes in the changes after prolonged stimulation, but not proteins associated with the BBB or inflammation. Next, they address the application to humans using a squeeze ball task. They imaged the brain and suggested that the hand activity led to an increased permeability of the vessels, suggesting modulation of the BBB.

Strengths:

The strengths of the paper are the novelty of the idea that stimulation of the limb can induce cortical plasticity in a normal condition, and it involves the opening of the BBB with albumin entry. In addition, there are many datasets, both rat and human data.

Weaknesses:

The explanation of why prolonged stimulation in the rat was considered relevant to normal conditions is still somewhat weak. The authors argue that the stimulation frequency they used is similar to rhythmic whisker movement. That is a good argument. However, the intensity they used, 2 mA is in the range they say can elicit a seizure if stimulation is 50 Hz. So that weakens the argument.

The authors made a lot of the requested changes but some questions were not addressed or the explanations were so brief that the confusion remained. Please go over the revisions again and make sure sentences are complete, jargon is explained, and arguments/justifications are clear. It will help the reader greatly.

The authors responded to the previous comments of Reviewer 2 regarding experimental design and variability of washout periods. It would be useful to incorporate the response into the paper so the readers know why the authors think the variability was not an important factor in the results.

Comments on the revised version:

The manuscript is improved.

---

## [Author Response]

The following is the authors’ response to the previous reviews

**Public Reviews:**

**Reviewer #1 (Public Review):**
The goal of the current study was to evaluate the effect of neuronal activity on blood-brain barrier permeability in the healthy brain, and to determine whether changes in BBB dynamics play a role in cortical plasticity. The authors used a variety of well-validated approaches to first demonstrate that limb stimulation increases BBB permeability. Using in vivo-electrophysiology and pharmacological approaches, the authors demonstrate that albumin is sufficient to induce cortical potentiation and that BBB transporters are necessary for stimulus-induced potentiation. The authors include a transcriptional analysis and differential expression of genes associated with plasticity, TGF-beta signaling, and extracellular matrix were observed following stimulation. Overall, the results obtained in rodents are compelling and support the authors' conclusions that neuronal activity modulates the BBB in the healthy brain and that mechanisms downstream of BBB permeability changes play a role in stimulus-evoked plasticity. These findings were further supported with fMRI and BBB permeability measurements performed in healthy human subjects performing a simple sensorimotor task. There is literature to suggest that there are sex differences in BBB dysfunction in pathophysiological conditions and the authors have acknowledged the use of only males as a minor limitation of the study that should be addressed in the future. Future studies should also test whether the upregulation of OAT3 plays a role in cortical plasticity observed following stimulation. Overall, this study provides novel insights into how neurovascular coupling, BBB permeability, and plasticity interact in the healthy brain.
**Reviewer #2 (Public Review):**
Summary:This study builds upon previous work that demonstrated that brain injury results in leakage of albumin across the blood brain barrier, resulting in activation of TGF-beta in astrocytes. Consequently, this leads to decreased glutamate uptake, reduced buffering of extracellular potassium and hyperexcitability. This study asks whether such a process can play a physiological role in cortical plasticity. They first show that stimulation of a forelimb for 30 minutes in a rat results in leakage of the blood brain barrier and extravasation of albumin on the contralateral but not ipsilateral cortex. The authors propose that the leakage is dependent upon neuronal excitability and is associated with an enhancement of excitatory transmission. Inhibiting the transport of albumin or the activation of TGF-beta prevents the enhancement of excitatory transmission. In addition, gene expression associated with TGF-beta activation, synaptic plasticity and extracellular matrix are enhanced on the "stimulated" hemisphere. That this may translate to humans is demonstrated by a break down in the blood brain barrier following activation of brain areas through a motor task.Strengths:This study is novel and the results are potentially important as they demonstrate an unexpected break down of the blood brain barrier with physiological activity and this may serve a physiological purpose, affecting synaptic plasticity.The strengths of the study are:(1) The use of an in vivo model with multiple methods to investigate the blood brain barrier response to a forelimb stimulation.(2) The determination of a potential functional role for the observed leakage of the blood brain barrier from both a genetic and electrophysiological view point(3) The demonstration that inhibiting different points in the putative pathway from activation of the cortex to transport of albumin and activation of the TGF-beta pathway, the effect on synaptic enhancement could be prevented. (4) Preliminary experiments demonstrating a similar observation of activity dependent break down of the blood brain barrier in humans.Weaknesses:The authors adequately addressed most of my points. A few remain:(1) Although the reviewers have addressed the possible effects of anaesthesia on neuro-vascular coupling. They have not mentioned or addressed the possible effects of ketamine (an NMDA receptor antagonist) on synaptic plasticity. Indeed, the low percentage of SEP increase following potentiation (10-20%) could perhaps be explained by partial block of NMDA receptors by ketamine.

We agree and apologize for this oversight. This important issue is now addressed in the Discussion.

“Notably, the antagonistic effect of ketamine on NMDA receptors might attenuate the magnitude of SEP potentiation recorded in our experiments (Anis et al., 1983; Salt et al., 1988).”

(2) The experimental paradigms remain unclear to me. Now, it appears that drugs are applied for 50 minutes and that the stimulation occurs during the "washout period". The more conventional approach would be to have the drug application during the stimulation period to determine if the drugs occlude or enhance the effects of stimulation and then washout the drugs. The problem is that drugs variably washout at different rates depending upon their lipid solubility.

We agree that the more conventional approach would have been to continue applying the drug throughout the experiment and that differential rates of washout may add variability to our experiments. However, despite this limitation, within each treatment group we found that the SEP response at 50 minutes (immediately after the drug application window) does not differ from SEP response at 80 minutes (after 30 minutes of stimulation and washout) [Figure 3H&G]. This suggests that the drug effects were still present despite terminating drug application and performing potentiation-inducing stimulation. Moreover, our analysis showed that animals within each treatment group (except AP5) had similar SEP responses with little intra-group variability.

(3) It is still not clear to what extent the experimenters and those doing the analysis were blinded to group. If one or both were blind to group, then please put this in the methods.

Thank you for this comment. We revised the Methods section to clearly confirm that data was collected and analyzed blindly.

**Reviewer #3 (Public Review):**
Summary:This study used prolonged stimulation of a limb to examine possible plasticity in somatosensory evoked potentials induced by the stimulation. They also studied the extent that the blood brain barrier (BBB) was opened by the prolonged stimulation and whether that played a role in the plasticity. They found that there was potentiation of the amplitude and area under the curve of the evoked potential after prolonged stimulation and this was long-lasting (>5 hrs). They also implicated extravasation of serum albumin, caveolae-mediated transcytosis, and TGFb signalling, as well as neuronal activity and upregulation of PSD95. Transcriptomics was done and implicated plasticity related genes in the changes after prolonged stimulation, but not proteins associated with the BBB or inflammation. Next, they address the application to humans using a squeeze ball task. They imaged the brain and suggest that the hand activity led to an increased permeability of the vessels, suggesting modulation of the BBB.Strengths:The strengths of the paper are the novelty of the idea that stimulation of the limb can induce cortical plasticity in a normal condition, and it involves opening of the BBB with albumin entry. In addition, there are many datasets and both rat and human data.Weaknesses:The conclusions are not compelling however because of a lack of explanation of methods.

In the revised paper, we added a section titled ‘study design’ that presents an overview of the experimental approach.

The explanation of why prolonged stimulation in the rat was considered relevant to normal conditions should be as clear in the paper as it is in the rebuttal.

We added a new paragraph to the Discussion section explaining this point as we did in the rebuttal:

“Our animal experiments show that a 30 min limb stimulation (at 6Hz and 2mA) increases cross-BBB influx, while a 1 min stimulation (of similar frequency and magnitude) does not. We believe that both types of stimulations fall within the physiological range because our continuous electrophysiological recordings showed no signs of epileptiform or otherwise pathological activity. Moreover, the recorded SEP levels were similar to those reported in previous physiological LTP studies in rats (Eckert & Abraham, 2010; Han et al., 2015; Mégevand et al., 2009) and humans (McGregor et al., 2016). In humans, skill acquisition often involves motor training sessions that last ≥30 minutes (Bengtsson et al., 2005; Classen et al., 1998) and result in physiological plasticity of sensory and motor systems (Classen et al., 1998; Draganski et al., 2004; Sagi et al., 2012). Hence, the experimental task in our human study (30 minutes of repetitive squeezing of an elastic stress-ball) is likely to represent physiological activity, with neuronal activation in primarily motor and sensory areas (Halder et al., 2005). Future human and animal studies are needed to explore the BBB modulating effects of additional stimulation protocols – with varying durations, frequencies, and magnitudes. Such studies may also elucidate the temporal and ultrastructural characteristics that differentiate between physiological and pathological BBB modulation. “

The authors need to ensure other aspects of the rebuttal are as clear in the paper as in the rebuttal too.

Thank you for this comment. This was addressed in the revised paper.

The only remaining concern that is significant is that it is hard to understand the figures.

Thank you for this comment. We revised the figures according to the reviewer’s recommendations. We hope that these changes increase the legibility of the figures.

**Reviewer #3 (Recommendations For The Authors):**
The manuscript is improved but there are still suggestions that do not appear to have been addressed. More experiments are not involved in addressing these concerns but one wants the paper to be clarified in terms of what was done.Figures. Please use arrows to point to the effect that the reader should see. Please note what the main point is.Major concerns:Please add explanations, exact p values, and other revisions in the rebuttal to the paper.

Rebuttal explanations were added to the paper and p values appear in figure legends.

Fig 1d shows a seizure-like event which the authors don't think is a seizure because it lacks a depolarization ship. This explanation is not convincing because a LFP would not necessarily show a depolarization ship. Another argument of a discussion of the event as a seizure is warranted. Note that expanding the trace might also show it is unlike a seizure. Regarding the idea that 6Hz 2 mA stimuli for 30 min are physiological, the authors make three arguments which are not clear. First, no epileptiform activity was found, but in Fig. 1 it looks like a seizure occurred. Second, memory and skill acquisition in humans open involve a similar training duration - but what about 6Hz 2 mA?

Rats are known to rhythmically move their whiskers at frequencies ranging between 5 and 15 Hz (Mégevand et al., 2009). We agree that there is no clear way to justify the similarity between the experimental design in humans and rats. However, we believe that both paradigms (paw stimulation in rats and ball squeeze in humans) represent non-pathological input that we found to modulate barrier permeability. This argument was added to the discussion of the paper:

“We believe that both types of stimulations fall within the physiological range because in rats, activity between 515 Hz represents physiological rhythmic whisker movement during environment exploration (Mégevand et al., 2009).”

Seizures are typically induced in rats via direct tetanic stimulation of the brain (at 50 Hz and 0.3-2.5mA) or maximal electroshock test to the cornea (at 50 Hz and 150 mA) (Swinyard et al., 1952). We, therefore, assert that the activity we observe represents physiological responses and not seizures. This argument is beyond the scope of the current paper.

Please note a limitation is that the high level of serum albumin is unlikely to be physiological but may not have been as high in the animal because of the low diffusion rate and degradation (please add the refs in the rebuttal).

Thank you, we added the following to the Results section:

“The relatively high concentration of albumin was chosen to account for factors that lower its effective tissue concentration such as its low diffusion rate and its likelihood to encounter a degradation site or a cross-BBB efflux transporter (Tao & Nicholson, 1996; Zhang & Pardridge, 2001).”

Fig. 1.Please consider a box in b to show where the expanded traces in the lower row came from.

Thank you for the suggestion. We added lines indicating where the trace excerpts were taken from.

c. Please use arrows to point to the parts that the authors want the reader to note. In the legend, explain what t is, and delta HbT.

Thank you. We implemented this suggestion.

d. It is not clear what the double-sided arrows are meant to show compared to the arrow without two sides.

We replaced the two-headed arrow with two single ones.

e. Please explain what the upward lines at the top signify. What does the red asterisk mean?

Thank you. We implemented this suggestion.

f. Is the reader supposed to note the yellow area? Please make it with an arrow or circle if so.

Thank you, we added a white circle to mark the area of tracer accumulation.

g. Please explain what the permeability index is or reference the part of the paper that does.

Further to this suggestion, we added a refence to the appropriate methods section to the legend.

h. Please use arrows to point to the area of interest.

Thank you. We implemented this suggestion.

m-n. Please mark areas of interest with arrows. m. the top right two images are unclear. I suggest making them say ipsi inset and contra inset instead of using asterisks.

Thank you. We added the ipsi and contra labels to panels in m. The images in panel n represent a phenomenon with no particular region of interest, but rather peri-vascular tracer accumulation along the entire depicted blood vessel. We clarified that panel n represents a separate experiment than panel m: “n. In an animal injected with both EB and NaFlu post stimulation, fluorescence imaging shows extravascular accumulation of both tracers along a cortical small vessel in the stimulated hemisphere.”

Figure 2.(2) a. Middle. What are the vertical lines at the top? The rebuttal states that was explained in the revised legends but I don't see it.

Our apologies. We now included an explanation that “an excerpt of the stimulation trace is shown above the middle LFP trace”.

c and d are very different field potentials in shape and therefore hard to compare. The rebuttal addresses this but the explanation is not in the revised text.

We agree that there is variability in SEP responses between animals. We now added a statement acknowledging this in the methods section: “To overcome potential variability in SEP morphology between animals (Mégevand et al., 2009), each animal’s plasticity measures (max amplitude and AUC of post stimulation SEP) were compared to the same measures at baseline.”

In d, it is not clear there is potentiation because the traces are not aligned.

All panels depicting SEP traces represent raw data with no alignment. The shift observed in panel d exemplifies why we compare post-stimulation parameters of max amplitude and area under curve to baseline in each animal.

Exact P values are said to have been added in the rebuttal but they were not.

Exact P values appear in Figure legends.

(3) b. Use arrows to mark the area of interest.

Thank you. We added a white circle to mark the area of tracer accumulation similar to Figure 1f.

d. Why is there an oscillation superimposed on all traces except CNQX?

We agree that this is an interesting question. Future studies should determine the source of this SEP pattern.

(4) What does the line and the number 2 mean? How were data normalized? What was counted? What area of cortex?

The number 2 refers to the scale bar line, meaning a log fold change of 2 reflects the size of the scale bar line.

The plot shows the log fold change against the mean count of each gene in the contralateral somatosensory cortex between 1 and 24 hours after stimulation.

The x axis title was changed to “mean expression” and the legend was modified to:

“Scatter plot of gene expression from RNA-seq in the contralateral somatosensory cortex 24 vs. 1 h after 30 min stimulation. The y axis represents the log fold change, and the x axis represents the mean expression levels (see methods, RNA Sequencing & Bioinformatics). Blue dots indicate statistically significant differentially expressed genes (DEGs) by Wald Test (n = 8 rats per group).”

How were the pericytes, smooth muscle cells, ,etc. distinguished?

This was explained under Methods->RNA Sequencing & Bioinformatics: “Analysis of cell-specific and vascular zonation genes was performed as described (Vanlandewijck et al., 2018), using the database provided in (http://betsholtzlab.org/VascularSingleCells/database.html)”.

What were the chi square statistics? If there were cells used instead of rats, please justify.

Thank you. The legend was expanded to include the following:

“The contralateral somatosensory cortex was found to have a significantly higher number of DEGs related to synaptic plasticity, than the ipsilateral side (***p<0.001, Chi-square).”

(5) b. what do the icons mean?

We agree that the icons were confusing. We simplified this panel to just show when participants were asked to squeeze the ball (black icon). This explanation was added to the Figure legend.

Abbreviations?

Abbreviations of MRI protocols were added to the figure legend for clarity.

In c-e what are the units of measure? Fold-change?

The units represent t-statistics values for each voxel. The label ‘t-statistic’ was added to the figure.

What are the white Iines, + and - signs?

The white lines point to voxels of highest activation (t-statistic). This was added to the legend.

And these are not +/- signs these are voxels with significant activation which only appear similar.

f. Please explain f and g for clarity.

Thank you. The explanation was modified for added clarity.

Supplemental Fig. 4.Original question: If ipsilateral and contralateral showed many changes why do the authors think the effects were only contralateral?The authors replied: Our gene analysis was designed to complement our in vivo and histological findings, by assessing the magnitude of change in differentially expressed genes (DEGs). This analysis showed that: (1) the hemisphere contralateral to the stimulus has significantly more DEGs than the ipsilateral hemisphere; and (2) the DEGs were related to synaptic plasticity and TGF-b signaling. These findings strengthen the hypothesis raised by our in vivo and histological experiments.Could the authors clarify the answer to the question in the text?

Thank you. This section was added to the Discussion.

Papers referenced in this letter:

Anis, N. A., Berry, S. C., Burton, N. R., & Lodge, D. (1983). The dissociative anaesthetics, ketamine and phencyclidine, selectively reduce excitation of central mammalian neurones by N-methyl-aspartate. British Journal of Pharmacology, 79(2), 565–575. hQps://doi.org/10.1111/j.1476-5381.1983.tb11031.x

Bengtsson, S. L., Nagy, Z., Skare, S., Forsman, L., Forssberg, H., & Ullén, F. (2005). Extensive piano practicing has regionally specific effects on white matter development. Nature Neuroscience, 8(9), 1148–1150. hQps://doi.org/10.1038/nn1516

Classen, J., Liepert, J., Wise, S. P., Hallett, M., & Cohen, L. G. (1998). Rapid plasticity of human cortical movement representation induced by practice. Journal of Neurophysiology, 79(2), 1117–1123. hQps://doi.org/10.1152/JN.1998.79.2.1117/ASSET/IMAGES/LARGE/JNP.JA47F4.JPEG

Draganski, B., Gaser, C., Busch, V., Schuierer, G., Bogdahn, U., & May, A. (2004). Changes in grey matter induced by training. Nature, 427(6972), 311–312. hQps://doi.org/10.1038/427311a

Eckert, M. J., & Abraham, W. C. (2010). Physiological effects of enriched environment exposure and LTP induction in the hippocampus in vivo do not transfer faithfully to in vitro slices. Learning and Memory, 17(10), 480–484. hQps://doi.org/10.1101/lm.1822610

Halder, P., Sterr, A., Brem, S., Bucher, K., Kollias, S., & Brandeis, D. (2005). Electrophysiological evidence for cortical plasticity with movement repetition. European Journal of Neuroscience, 21(8), 2271–2277. hQps://doi.org/10.1111/J.1460-9568.2005.04045.X

Han, Y., Huang, M. De, Sun, M. L., Duan, S., & Yu, Y. Q. (2015). Long-term synaptic plasticity in rat barrel cortex. Cerebral Cortex, 25(9), 2741–2751. hQps://doi.org/10.1093/cercor/bhu071

McGregor, H. R., Cashaback, J. G. A., & Gribble, P. L. (2016). Functional Plasticity in Somatosensory Cortex Supports Motor Learning by Observing. Current Biology, 26(7), 921–927. hQps://doi.org/10.1016/j.cub.2016.01.064

Mégevand, P., Troncoso, E., Quairiaux, C., Muller, D., Michel, C. M., & Kiss, J. Z. (2009). Long-term plasticity in mouse sensorimotor circuits after rhythmic whisker stimulation. Journal of Neuroscience, 29(16), 5326– 5335. hQps://doi.org/10.1523/JNEUROSCI.5965-08.2009

Sagi, Y., Tavor, I., HofsteQer, S., Tzur-Moryosef, S., Blumenfeld-Katzir, T., & Assaf, Y. (2012). Learning in the Fast Lane: New Insights into Neuroplasticity. Neuron, 73(6), 1195–1203. hQps://doi.org/10.1016/j.neuron.2012.01.025

Salt, T. E., Wilson, D. G., & Prasad, S. K. (1988). Antagonism of N-methylaspartate and synapBc responses of neurones in the rat ventrobasal thalamus by ketamine and MK-801. British Journal of Pharmacology,

94(2), 443–448. hQps://doi.org/10.1111/j.1476-5381.1988.tb11546.x

Swinyard, E. A., Brown, W. C., & Goodman, L. S. (1952). Comparative assays of antiepileptic drugs in mice and rats. The Journal of Pharmacology and Experimental Therapeutics, 106(3), 319–330. hQp://jpet.aspetjournals.org/content/106/3/319.abstract

Tao, L., & Nicholson, C. (1996). Diffusion of albumins in rat cortical slices and relevance to volume transmission. Neuroscience, 75(3), 839–847. hQps://doi.org/10.1016/0306-4522(96)00303-X

Vanlandewijck, M., He, L., Mäe, M. A., Andrae, J., Ando, K., Del Gaudio, F., Nahar, K., Lebouvier, T., Laviña, B.,

Gouveia, L., Sun, Y., Raschperger, E., Räsänen, M., Zarb, Y., Mochizuki, N., Keller, A., Lendahl, U., &

Betsholtz, C. (2018). A molecular atlas of cell types and zonation in the brain vasculature. Nature, 554(7693), 475–480. hQps://doi.org/10.1038/nature25739

Zhang, Y., & Pardridge, W. M. (2001). Mediated efflux of IgG molecules from brain to blood across the blood– brain barrier. Journal of Neuroimmunology, 114(1–2), 168–172. hQps://doi.org/10.1016/S01655728(01)00242-9